# Chest CT findings related to mortality of patients with COVID-19: A retrospective case-series study

Yiqi Hu[1]☯, Chenao Zhan[1]☯, Chengyang Chen[2], Tao Ai[1]*, Liming Xia[1]

1 Department of Radiology, Tongji Hospital, Tongji Medical College, Huazhong University of Science and Technology, Wuhan, China, 2 Department of Radiology, The First Hospital of Hebei Medical University, Shijiazhuang, China

☯ These authors contributed equally to this work.
* aitao007@hotmail.com

**Data Availability Statement:** All relevant data are within the manuscript.

**Funding:** The author(s) received no specific funding for this work.

## Abstract

### Background

As the current outbreak of COVID-2019 disease has spread to the other more than 150 countries besides China around the world and the death number constantly increased, the clinical data and radiological findings of death cases need to be explored so that more physicians, radiologists and researchers can gain important information to save more lives.

### Methods

73 patients who died from COVID-19 were retrospectively included. The clinical and laboratory data of the patients were extracted from electronic medical records. The clinical data, inflammation-related laboratory results, and CT imaging features were summarized. The laboratory results and dynamic changes of imaging features and severity scores of lung involvement based on chest CT were analyzed.

### Results

The mean age was 67±12 years. The typical clinical symptoms included fever (88%), cough (62%) and dyspnea (23%). 65% patients had at least one underlying disease. GGO with consolidation was the most common feature for the five lung lobes (47%-53% among the various lobes), with total severity score of 12.97±5.87 for the both lungs. The proportion of GGO with consolidation is markedly increased on follow-up chest CT compared with initial CT scans, as well as the averaging total CT scores (14.53±5.76 vs. 6.60±5.65; P<0.001). The severity score was rated as severe (white lung) in 13% patients on initial CT scans, and in 60% on follow-up CT scans. Moderate positive correlations were found between CT scores and leucocytes, neutrophils and IL-2R (r = 0.447–0581, P<0.001).

### Conclusion

Chest CT findings and laboratory test results were worsening in patients who died of COVID-19, with moderate positive correlations between CT severity scores and

**Competing interests:** The authors have declared that no competing interests exist.

inflammation-related factors of leucocytes, neutrophils, and IL-2R. Chest CT imaging may play an more important role in monitoring disease progression and predicting prognosis.

## Introduction

In December 2019, an outbreak of new coronavirus disease (COVID-19) caused by severe acute respiratory syndrome coronavirus 2 (SARS-CoV-2) was initially reported in Wuhan City, China. The disease was then rapidly sweeping through the whole country and has spread to the other more than 150 countries and territories around the world. On March 13, 2020, the World Health Organization (WHO) declared the novel coronavirus outbreak to be a pandemic [1] and as of July 12, 2020, 7:15 GMT, accumulative 12,879,917 confirmed cases and 568,546 deaths were reported in countries.

The infection of SARS-CoV-2 shares highly homological features with SARS-CoV, and causes acute, lethal respiratory pneumonia with typical clinical symptoms of fever and cough. According to an earlier report by WHO, most patients (80%) experienced mild to moderate illness, and about 14% experienced severe disease and 5% were critically ill [2, 3].

The estimated mortality rate of COVID-19 is 3.5% on a global-wide scale, which is relatively low compared with the SARS-CoV and MERS-CoV [4]. However, it varies by location, the intensity of transmission, and infection prevention and control measures. The mortality reached up to 6% in Wuhan (early epicenter area) in the early stage of the outbreak, and current 12.7% in Italy due to the rapid increase in the number of infections and shortage of medical resources. What is more, the mortality in critically ill patients has been reported as high as 60% [5]. In this context, early identification of risk factors for poor prognosis, accurate evaluation of disease severity and monitoring disease progression will be essential to reduce the mortality rate of patients with COVID-19.

Chest CT imaging plays a valuable role in the screening and dynamic evaluation of patients with COVID-19 [6, 7]. Most patients with COVID-19 have typical CT imaging features of multiple ground-glass opacity (GGO) and/or consolidations in a peripheral distribution, which also reflects the severity of pulmonary inflammation. We assumed that chest CT features and dynamic changes may serve as an important biomarker for the risk stratification, prognostic prediction, and therapy decision of severe patients with COVID-19. To the best of our knowledge, no studies regarding chest CT characteristics related to the mortality of patients with COVID-19 have been reported until now.

In this study, we aimed to explore the chest CT imaging features of patients who died from COVID-19, correlating with essential clinical information and laboratory test results. We hope the results of this study will contribute to clinicians' comprehension and treatment plan of COVID-19.

## Materials and methods

### Patients

This retrospective study was approved by the institutional review boards (IRB) of Tongji Hospital and written informed consent was waived due to the outbreak of COVID-19. 153 patients who highly likely died from COVID-19 from January 27 to February 28, 2020, were retrospectively reviewed in Tongji hospital (the largest general hospital designated for the treatment of severe patients with COVID-19 in Wuhan). The inclusion criteria based on the latest guideline of Diagnosis and Treatment of Pneumonitis caused by COVID-2019 [8] were: (1) positive

real-time reverse transcriptase-polymerase chain reaction (rRT-PCR) results with throat swab or lower respiratory tract samples; (2) available chest CT images; (3) available clinical and laboratory data. The exclusion criteria were: (1) the patient had chest CT images with very poor quality due to motion artifact; (2) the patient died of a severe underlying extra pulmonary disease. Finally, 73 patients were included in this study.

## Clinical and laboratory data

The clinical and laboratory data of the patients were extracted from clinical electronic medical records in the hospital information system (HIS). The demographic information, clinical symptoms and signs, underlying chronic diseases were recorded for all patients. Laboratory results mainly included peripheral blood cell counts, inflammatory cytokines (interleukin-1β [IL-1β], interleukin-2 receptor [IL-2R], interleukin-6 [IL-6], interleukin-8 [IL-8], interleukin-10 [IL-10], and tumor necrosis factor α [TNF-α]), and infection-related biomarkers (procalcitonin [PCT], erythrocyte sedimentation rate [ESR], serum ferritin [SF], high sensitive C-reactive protein [hs-CRP]). The results of laboratory tests immediately after the hospital admission and before the death were selected for the comparison in this study.

## Chest CT acquisition

All chest scans images were obtained with three CT systems (uCT 780, United Imaging, China; Optima 660, GE, America; Somatom Definition AS+, Siemens Healthineers, Germany). The patients were scanned in a supine position during breath-holding. The main imaging parameters were: tube voltage = 120 kVp, automatic tube current modulation (30–70 mAs), pitch = 0.99–1.22 mm, matrix = 512 × 512, slice thickness = 10 mm, field of view = 350 mm × 350 mm. All images were then reconstructed with a slice interval of 0.625 to 1.250 mm.

## CT image analysis

All CT images were analyzed by two chest radiologists in consensus (Y.Q.H. and C.A.Z with 6 and 3 years of experience in interpreting chest CT images, respectively), who were blinded to clinical and laboratory data. The main features of CT images were described as the following four patterns: ground-glass opacity, ground-glass opacity with consolidation, consolidation and other (linear opacities, traction bronchiectasis, cysts, and reticular opacities) for each patient's initial chest CT scan and last follow-ups if available. Each of the five lung lobes was visually scored for the degree of lung involvements using a 4-point- scale: 0, no involvement; 1, 1–25% involvement; 2, 26%-49% involvement; 3, 50%-75% involvement; 4, 76%-100% involvement [9]. The total severity CT score (the extent of pulmonary disease) was the sum of the five individual lobar scores and defined as follows: 0, none; 1–5, minimal; 6–10, mild; 11–15, moderate; and 16–20, severe involvement of the lung (white lung). The dynamic changes of imaging features and total severity scores were evaluated on the follow-up chest CT scans if available.

## Statistical analysis

Statistical analysis was performed by using SPSS for Windows (version 20.0, SPSS Inc., Chicago, IL, USA). All quantitative variables were expressed as mean value ± standard deviation (SD) or median with range unless otherwise specified. All categorical variables were expressed as frequency with percentage. The differences in laboratory test results immediately after the hospital admission and before the death were compared by the paired-sample t-test. The difference in chest CT scores between the initial and follow-up CT scans, and among the different

periods were compared by Wilcoxon rank-sum test and Mann-Whitney U test. The correlation between CT scores and inflammation-related parameters (selected from the time point closest to the examination time of chest CT scans) was calculated by Spearman correlation. The strength of the correlation (r) was defined as follows: $|r| < 0.20$, very weak; $0.20 \leq |r| < 0.40$, weak; $0.40 \leq |r| < 0.60$, moderate; $0.60 \leq |r| < 0.80$, strong; $0.80 \leq |r| \leq 1.0$, very strong. A P-value of less than 0.05 was considered statistically significant.

## Results

### General clinical and laboratory results

A total of 73 patients with confirmed COVID-19 were enrolled in this retrospective study. The mean age was 67±12 years with a range of 33–95 years. 65% patients had at least one underlying disease or condition (Table 1). 9% patients had chronic pulmonary diseases and 10% patients had a smoking history. The typical clinical symptoms included fever (88%), dry cough (62%) and dyspnea (23%). Besides, 10% patients experienced diarrhea during the disease course. 26% patients received intubation. 1% patients underwent a tracheotomy and 3% patients received extracorporeal membrane oxygenation (ECMO) therapy. For the laboratory tests (Table 2), leucocytes, neutrophils and neutrophil-to-lymphocyte ratio at the time before the death were significantly elevated than those at the time immediately after the admission (P<0.001, P = 0.004, 0.014, respectively). The levels of inflammatory cytokines of IL-6 and TNF-α and infection-related biomarker of SF were also significantly increased with the disease progression (P = 0.038, 0.048 and 0.022, respectively).

### CT imaging features and dynamic changes

Of 73 patients, 58 patients had only one chest CT scan; 11 had two chest CT scans; three had three chest CT scans; one had four chest CT scans during the disease course (total 93 CT scans). The median time interval between the initial chest CT scans and the last follow-up chest CT scans was five days with a range of 2 to 27 days. The last CT scan was selected to summarize the imaging features for patients with follow-up CT scan(s). Ground-glass opacity with consolidation was the most common feature in each of the five lung lobes (47%-53% among the various lobes; Table 3), followed by GGO in three lobes of the right lung and left upper lobe (16% to 29%), and consolidation in left lower lobe (19%). The most affected lobes were lower lobes of each side of the lung (CT scores of 2.97±1.34 and 2.79±1.34, respectively). The total severity score was 12.97±5.87. The total scores were rated as moderate (11–15 points) and severe (16–20 points) in 22% and 47% patients, respectively.

As compared with the initial chest CT scans, the proportion of GGO with consolidation is markedly increased on the follow-up chest CT scans (Table 4). The averages CT scores were significantly increased on follow-up CT scans than those on initial CT scans for each of the five lobes, as well as the average CT score for both lungs (14.53±5.76 vs. 6.60±5.65; P<0.001). The severity score was rated as severe (white lung) in 13% (2/15) patients based on the initial CT scans, and in 60% (9/15) patients based on the follow-up CT scans (Figs 1–3).

When tracking back the different periods (<7 days, 7–14 days, >14 days) before the death, the average chest CT scores were 14.93±5.06, 14.09±4.67 and 9.57±6.93, respectively (Table 5). A significant difference was found between the score based on chest CT scans greater than 14 days and those within 14 days before the death (P = 0.004 and 0.013, respectively). The severity scores were rated as severe (white lung) in 57% (17/30), 51% (18/35) and 32% chest CT scans for each period before the death (from near to far).

**Table 1. Clinical characteristics of all patients.**

|  | Patient (n = 73) |
|---|---|
| **Age (years)** |  |
| Mean | 67 ± 12 |
| Range | 33–95 |
| ≤40 | 1 (1%) |
| 41–60 | 20 (27%) |
| 61–80 | 41 (56%) |
| ≥81 | 11 (15%) |
| **Male** | 54 (74%) |
| **Underlying disease or conditions** | 44/68 (65%) |
| Hypertension | 29/68 (43%) |
| Diabetes | 12/68 (18%) |
| Cardiovascular/Cerebrovascular disease | 12/68 (18%) |
| Chronic kidney disease | 7/68 (10%) |
| Malignancy | 7/68 (10%) |
| Chronic pulmonary disease | 6/68 (9%) |
| Administration of immunosuppressive agents | 1/68 (1%) |
| **Smoking history** | 7/73 (10%) |
| **Initial CT scan to (median, days)** |  |
| Onset of symptoms | 8 (1–39) |
| Death | 11 (1–36) |
| Follow up CT | 5 (2–27) |
| **Onset of symptoms to death (median, days)** | 19 (8–41) |
| **Symptoms and signs** |  |
| Fever | 64/73 (88%) |
| Dry cough | 45/73 (62%) |
| Expectoration | 10/73 (14%) |
| Dyspnea | 17/73 (23%) |
| Chest distress | 16/73 (22%) |
| Fatigue | 13/73 (18%) |
| Diarrhea | 7/73 (10%) |
| Chills | 2/73 (3%) |
| Myalgia | 3/73 (4%) |
| Dizziness | 2/73 (3%) |
| Coma | 1/73 (1%) |
| None | 1/73 (1%) |
| **Intubation** | 19/73 (26%) |
| **Tracheotomy** | 1/73 (1%) |
| **ECMO** | 2/73 (3%) |

Value listed as mean N (%), except annotated values.

## Correlation of CT scores and inflammatory parameters

Moderate positive correlations were found between mean CT scores and leucocytes (r = 0.581, P<0.001), neutrophils (r = 0.587, P<0.001), and IL-2R (r = 0.447, P = 0.003). Week positive correlations were found between mean CT scores and neutrophil-to-lymphocyte ratio

**Table 2. Laboratory test results of inflammation-related parameters.**

|  | After admission | Before death | P-value |
|---|---|---|---|
| **Leucocytes (×109 per L)** | 9.88 ± 5.26 | 12.53 ± 7.41 | <0.001 |
| Increased | 33/71 (46%) | 37/71 (52%) | |
| **Neutrophils (×109 per L)** | 8.68 ± 5.21 | 11.44 ± 12.07 | 0.003 |
| Increased | 46/71 (65%) | 51/71 (72%) | |
| **Lymphocytes (×109 per L)** | 0.71 ± 0.42 | 0.77 ± 1.40 | 0.739 |
| Decreased | 60/71 (85%) | 63/71 (89%) | |
| **Neutrophil-to-lymphocyte ratio** | 17.72 ± 15.66 | 27.14 ± 49.01 | 0.014 |
| **Interleukin-1β (pg/mL)** | 7.82 ± 12.72 | 7.72 ± 12.56 | 0.665 |
| Increased | 7/45 (16%) | 9/45 (20%) | |
| **Interleukin-2 receptor (U/mL)** | 1251.04 ± 791.74 | 1397.22 ± 903.77 | 0.109 |
| Increased | 35/45 (78%) | 35/45 (78%) | |
| **Interleukin-6 (pg/mL)** | 147.01 ± 315.60 | 425.45 ± 617.56 | 0.038 |
| Increased | 44/45 (98%) | 43/45 (96%) | |
| **Interleukin-8 (pg/mL)** | 50.85 ± 85.09 | 96.51 ± 246.52 | 0.179 |
| Increased | 10/45 (22%) | 14/45 (31%) | |
| **Interleukin-10 (pg/mL)** | 15.81 ± 18.12 | 17.95 ± 21.43 | 0.180 |
| Increased | 25/45 (56%) | 25/45 (56%) | |
| **Tumor necrosis factor α (pg/mL)** | 12.50 ± 6.96 | 15.85 ± 11.18 | 0.048 |
| Increased | 32/45 (71%) | 32/45 (71%) | |
| **Procalcitonin (ng/mL)** | 0.87 ± 1.78 | 1.53 ± 2.99 | 0.055 |
| Increased | 63/64 (98%) | 62/64 (97%) | |
| **Erythrocyte sedimentation rate (mm/H)** | 41.69 ± 29.25 | 40.46 ± 30.30 | 0.704 |
| Increased | 39/58 (67%) | 39/58 (67%) | |
| **Serum ferritin (ug/L)** | 1874.52 ± 1974.73 | 2409.30 ± 2330.41 | 0.022 |
| Increased | 37/38 (97%) | 37/38 (97%) | |
| **High sensitive C-reactive protein (mg/L)** | 112.35 ± 74.51 | 116.05 ± 84.99 | 0.703 |
| Increased | 63/70 (90%) | 63/70 (90%) | |

Value listed as mean±SD or N (%).

(r = 0.300, P = 0.012), and IL-8 (r = 0.332, P = 0.030). A week negative correlation was found between mean CT scores and lymphocytes (r = -0.286, P = 0.019; Fig 4).

## Discussion

With the rapid spread of COVID-19 around the world, it has stirred up an international concern. This study preliminarily demonstrated the features of chest CT imaging in severe

**Table 3. Image characteristics and severity scores of chest CT scans (N = 73).**

|  | GGO | GGO with Consolidation | Consolidation | Other | CT Score |
|---|---|---|---|---|---|
| Right upper lobe | 16 (22%) | 37 (51%) | 12 (16%) | 8 (11%) | 2.5 1± 1.38 |
| Right middle lobe | 21 (29%) | 34 (47%) | 9 (12%) | 9 (12%) | 2.32 ± 1.39 |
| Right lower lobe | 13 (18%) | 39 (53%) | 12 (16%) | 9 (12%) | 2.97 ± 1.34 |
| Left upper lobe | 16 (22%) | 34 (47%) | 11 (15%) | 11 (15%) | 2.41 ± 1.36 |
| Left lower lobe | 12 (16%) | 37 (51%) | 14 (19%) | 10 (14%) | 2.79 ± 1.34 |
| Total five lung lobes |  |  |  |  | 12.97 ± 5.87 |

Value listed as N (%) or mean ± SD.

**Table 4. Dynamic changes of imaging features and severity scores between the initial and follow-up CT scans (N = 15).**

| | GGO | | GGO with Consolidation | | Consolidation | | Other | | CT Score | | |
|---|---|---|---|---|---|---|---|---|---|---|---|
| | Initial | Follow-up | Initial | Follow-up | Initial | Follow-up | Initial | Follow-up | Initial | Follow-up | P-value |
| Right upper lobe | 2 (13%) | 2 (13%) | 5 (33%) | 12 (80%) | 3 (20%) | 1 (7%) | 5 (33%) | 0 (0%) | 1.13 ± 1.13 | 3.07 ± 1.16 | <0.001 |
| Right middle lobe | 4 (27%) | 3 (20%) | 4 (27%) | 10 (67%) | 1 (7%) | 1 (7%) | 6 (40%) | 1 (7%) | 1.20 ± 1.21 | 2.73 ± 1.39 | <0.001 |
| Right lower lobe | 6 (40%) | 1 (7%) | 4 (27%) | 12 (80%) | 1 (7%) | 2 (13%) | 4 (27%) | 0 (0%) | 1.53 ± 1.46 | 3.27 ± 1.16 | <0.001 |
| Left upper lobe | 5 (33%) | 3 (20%) | 3 (20%) | 10 (67%) | 1 (7%) | 1 (7%) | 6 (40%) | 1 (7%) | 1.13 ± 1.13 | 2.53 ± 1.25 | <0.001 |
| Left lower lobe | 5 (33%) | 1 (7%) | 6 (40%) | 3 (20%) | 1 (7%) | 9 (60%) | 3 (20%) | 2 (13%) | 1.67 ± 1.23 | 2.93 ± 1.34 | <0.001 |
| Total five lung lobes | | | | | | | | | 6.60 ± 5.65 | 14.53 ± 5.76 | <0.001 |

Value listed as N (%) or mean ± SD.

patients died from COVID-19. By lung lobe-based analysis, the typical imaging features of GGO with consolidation was found in 47%-53% patients died from COVID-19, with an average severity score of 12.97±5.87. The proportions of consolidation and severity scores were significantly increased with the disease progression; meanwhile, the white lung was observed in more than half of the patients as early as 7–14 days before the death. In addition, moderate correlations between the severity of pulmonary inflammation based on chest CT imaging and inflammatory parameters (leukocytes, neutrophils, and IL-2R) were also firstly found in this study.

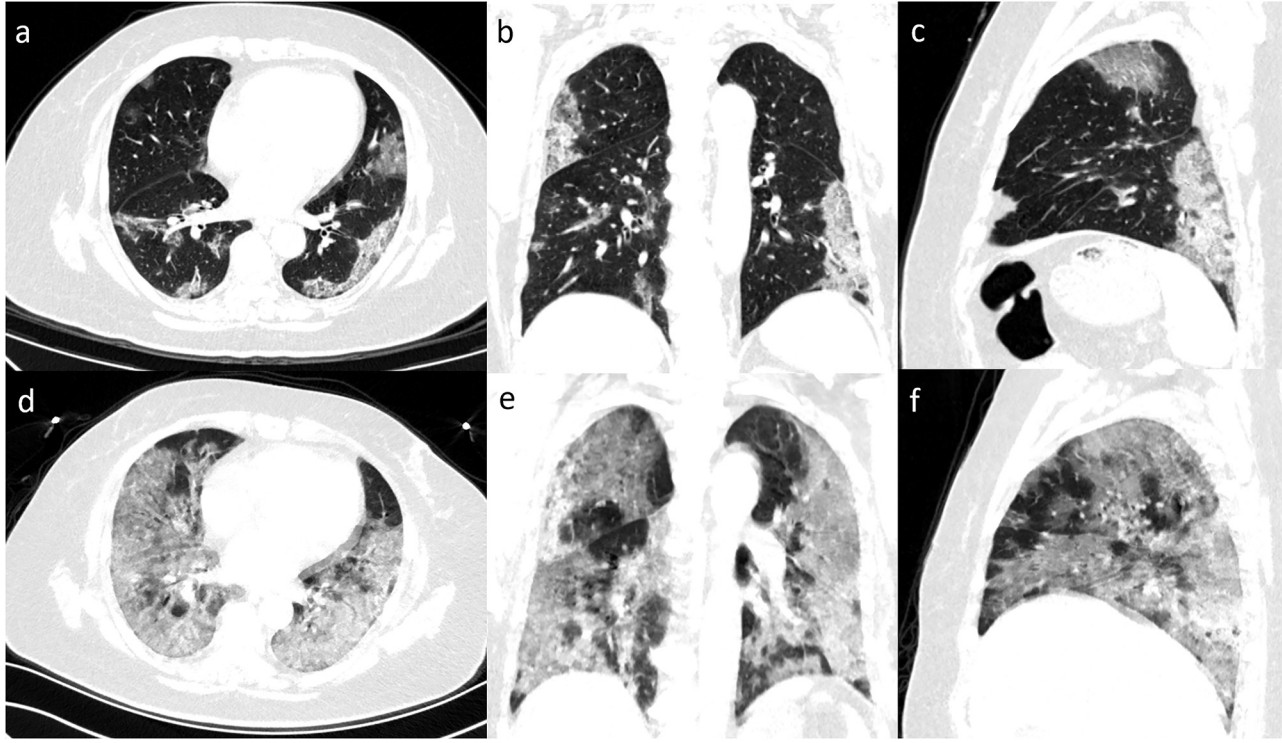

**Fig 1. Serial chest images of a 57-year-old man with fever and dry cough.** (a-d) 12 days before the death, chest CT with axial, coronal, sagittal planes showed moderate lung involvement with a total CT score of 11, with subpleural ground-glass opacities and patches of consolidation in bilateral lungs. (e-f) 4 days before the death, chest CT with different planes showed severe lung involvement with a total score of 20, with the appearance of the white lung.

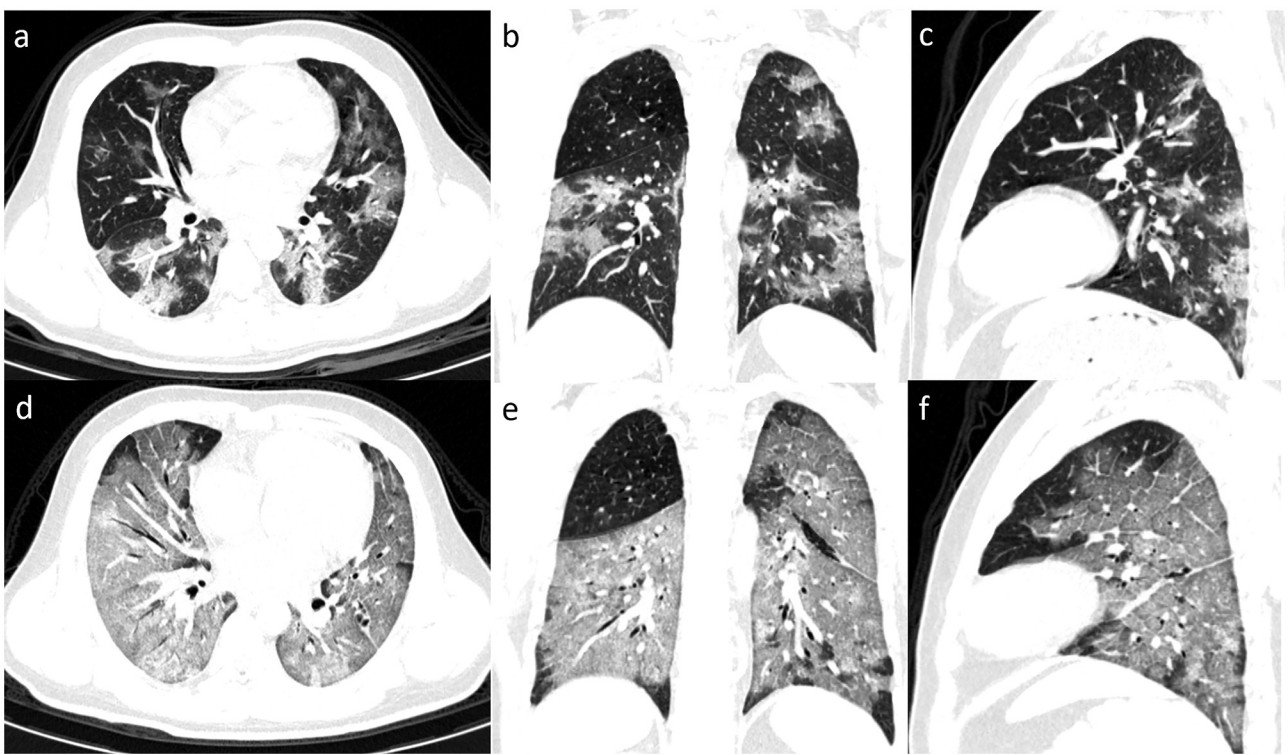

**Fig 2. Serial chest images of a 33-year-old man with fever and dyspnea.** (a-d) 16 days before death, chest CT with axial, coronal, sagittal planes showed moderate lung involvement with a total CT score of 11, appearing as ground-glass opacities (GGOs) with patches of consolidation in bilateral lungs. (e-f) 12 days before death, chest CT with different planes showed severe lung involvement with a total score of 19, appearing as diffusion GGOs with consolidation (white lung).

In this study, the average age of the patients was 67±12 years old; most of them had at least one underlying disease such as hypertension, diabetes, cardiovascular or cerebrovascular diseases. This is agreed with the previous report that the older patient with COVID-19 tends to become more severe, in particular for the patients with underlying diseases [10]. Fever and dry cough were the main symptoms after the onset of SARS-CoV-2 infection, which is similar to the symptoms of SARS or MERS [11]. For the laboratory, the significant increase in leukocytes, neutrophils and neutrophil-to-lymphocyte ratio suggested the severity of pulmonary inflammatory responses and impairment of the immune system in patients infected with SARS-CoV-2 [12]. The infection-related biomarkers including SF markedly elevated in the serum as well as Inflammatory cytokines including IL-6, and TNF-α, indicating that inflammation storm may also occur and aggravate in patients died from COVID-19 during the disease [13–17].

Chest CT imaging plays an important role in the diagnosis and dynamic evaluation of COVID-19. Typical imaging features of multiple ground-glass opacities and/or consolidations in patients with COVID-19 pneumonia have been detailedly described in previous reports [18, 19]. Even though the pathogenesis of SARS-CoV-2 infection is not fully understood, diffuse alveolar injury and progressive respiratory failure caused by SARS-CoV-2 is the leading cause of death in severe patients with COVID-19 [20]. From the current study, we found GGO and GGO with consolidation were the most predominant imaging features in patients who died from COVID-19, which is correlated with the pathological findings of COVID-19 that severe inflammatory exudation in intra-alveolar spaces and hyaline membrane formation [20, 21].

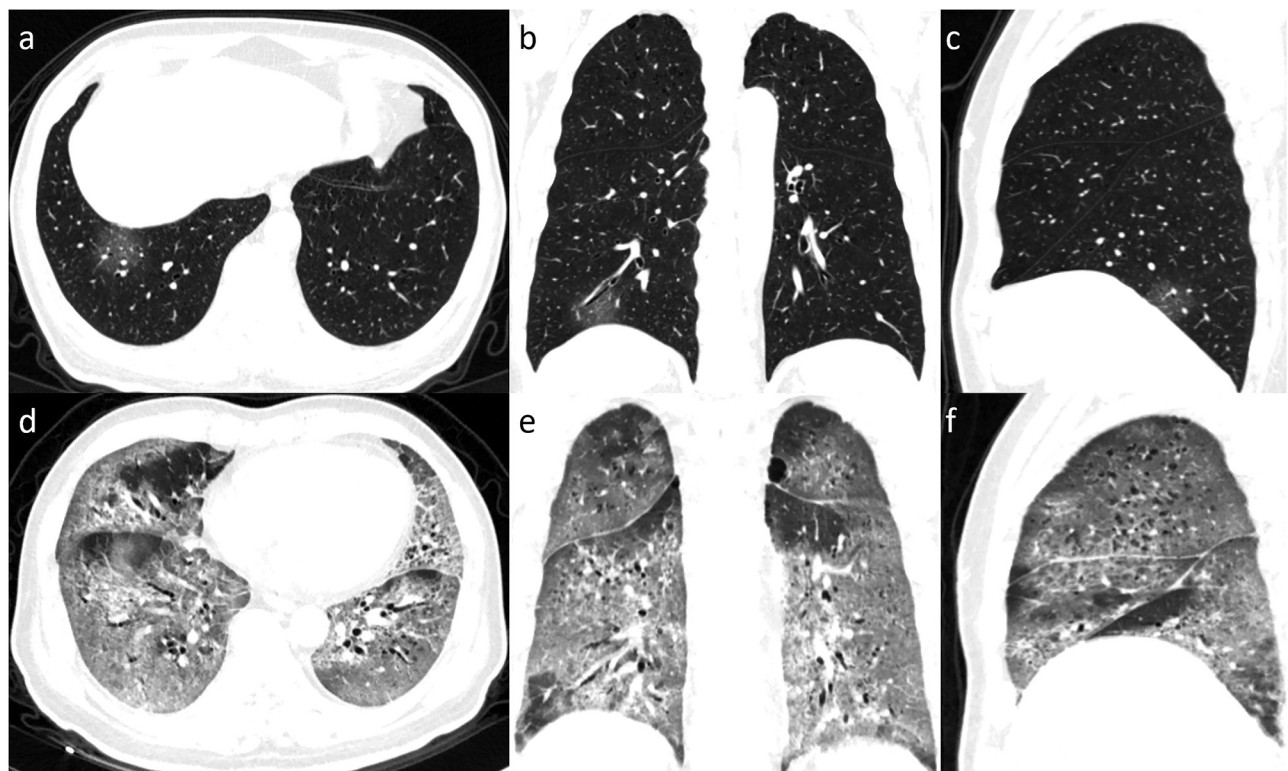

**Fig 3. Serial chest images of a 59-year-old man with fever, dyspnea, and dry cough.** (a-d) 17 days before death, chest CT with axial, coronal, sagittal planes showed minimal lung involvement with a total CT score of 1, a small piece of ground-glass opacity was found in the right lower lobe. (e-f) 10 days before death, chest CT with different planes showed severe lung involvement with a total score of 19, with an appearance of the white lung.

The severity score of lung involvement in patients who died from COVID-19 was also significantly greater than that in patients with mild to moderate COVID-19 (12.97±5.87 vs. 7±4) [6]. What is more, the mild-moderate correlation between chest CT severity scores and systemic inflammation activation was also preliminarily demonstrated in this study. Therefore, the imaging features and dynamic changes could provide the most direct evidence for assessing the severity of the disease and the prognosis.

The results of this study also demonstrated that the proportions of consolidation and severity scores were significantly increased on the follow-up chest CT scans as compared with the initial CT scans (14.53 ± 5.76 vs. 6.60 ± 5.65) in patients died from COVID-19. More white

**Table 5. The mean CT scores (mean±sd) the number of patients/CT scans [n (%)] for the different severities of lung involvement between initial chest CT scans and follow-up chest CT scan (N = 15 patients), and among the different periods before the death (N = 93 CT scans).**

|  | n | Total Score | None | Minimal | Mild | Moderate | Severe |
|---|---|---|---|---|---|---|---|
| Initial CT | 15 | 6.60 ± 5.65 | 1 (7%) | 8 (53%) | 3 (20%) | 1 (7%) | 2 (13%) |
| Follow-up CT | 15 | 14.53 ± 5.76[a] | 0 (0) | 2 (13%) | 2 (13%) | 2 (13%) | 9 (60%) |
| 0–7 days | 30 | 14.93 ± 5.06 | 0 (0) | 3 (10%) | 3 (10%) | 7 (23%) | 17 (57%) |
| 7–14 days | 35 | 14.09 ± 4.67 | 0 (0) | 2 (6%) | 6 (17%) | 9 (26%) | 18 (51%) |
| > 14days | 28 | 9.57 ± 6.93[b] | 1 (4%) | 11 (39%) | 4 (14%) | 3 (11%) | 9 (32%) |

[a]. significant difference between initial and follow-up chest CT scans by Wilcoxon signed-rank test;

[b]. significant difference between the group with 14 days more and the groups with 0–7 and 7–14 days before the death by Mann-Whitney U test.

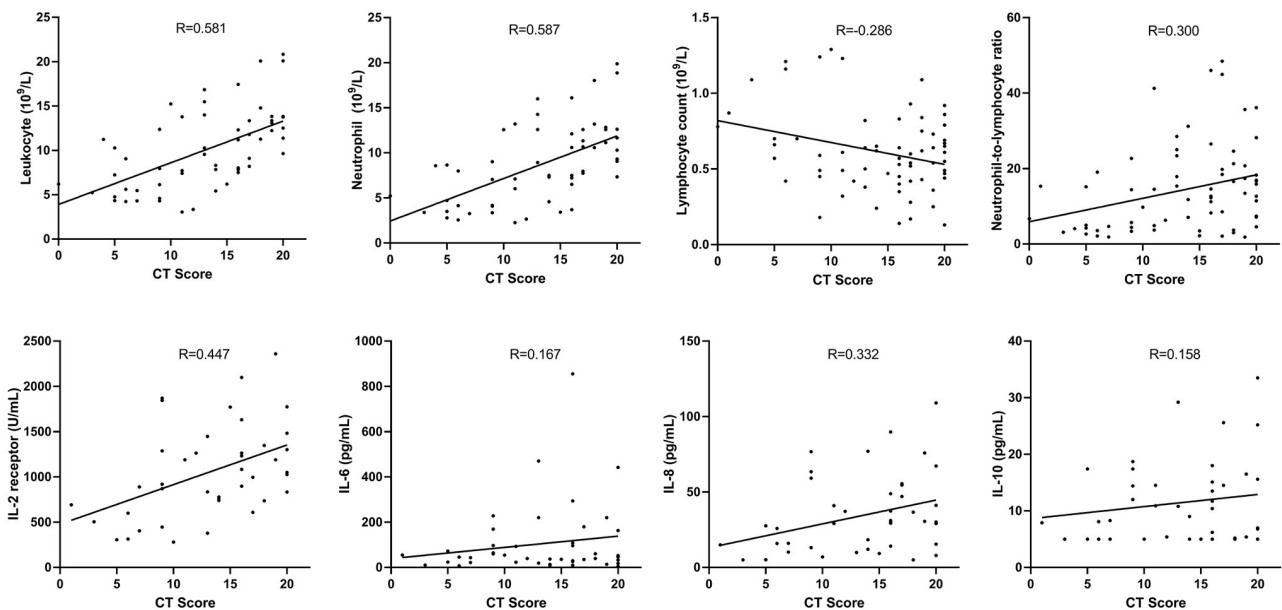

**Fig 4. The correlations of CT scores and inflammation-related parameters.**

lungs (severe lung involvement) were observed in the late stage of the disease (57% vs. 32% for 0–7 days and >14 days before the death). In addition, considering the correlations between chest CT scores (lung involvement) and inflammatory parameters of laboratory tests, the increase of the proportions of consolidation and the extent of GGO (higher CT scores) would suggest secondary bacterial infection occurred in the late stage of SARS-CoV-2 infection, which may be responsible for the rapid deterioration and acute respiratory distress in severe patients with COVID-19. Empirically administration of antibacterial drugs was thus recommended by some clinical experts. Therefore, rapidly increased consolidation and high severity scores based on chest CT images may predict the patient's poor prognosis.

The outbreak of COVID-19 has had a strong impact worldwide. Almost all countries have suffered huge losses in health, society and economy [22]. Our results may be potential risk factors to identify patients with poor prognosis, help clinicians to provide earlier interventions for these patients, and improve their survival rate.

There were some limitations in this present study. First, there was no control group included in this study. Thus, it is hard to evaluate the exact value of chest CT imaging in identifying the risk factors of poor prognosis as compared with the other clinical and/or laboratory parameters. A case-control study needs to be done shortly. Second, because many patients were transferred from the other hospitals, their early chest CT images were not available for us. In addition, some patients did not have follow-up chest CT scans due to critically ill conditions. As lung biopsy was lacked, the relationship between imaging features and histopathological findings needs to be investigated. Therefore, other potential causes of underlying disease were not estimated. Therefore, the information of dynamic changes of CT features and severity scores is limited.

## Conclusion

Chest CT findings and laboratory test results were worsening in patients who died of COVID-19, with moderate positive correlations between CT severity scores and inflammation-related factors of leucocytes, neutrophils, and IL-2R demonstrated in this study. Our results and

analysis suggest that chest CT features and severity scores with dynamic changes may serve as potential risk factors helping clinicians to identify patients with poor prognosis.

## Acknowledgments

Thanks W.Z.L. for the contribution to data collection.

## Author Contributions

**Conceptualization:** Tao Ai.

**Data curation:** Yiqi Hu, Chengyang Chen.

**Formal analysis:** Chenao Zhan.

**Software:** Chengyang Chen.

**Supervision:** Liming Xia.

**Validation:** Liming Xia.

**Writing – original draft:** Yiqi Hu.

**Writing – review & editing:** Tao Ai.

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
