## [Decision Letter · Decision Letter 0]

7 Jul 2020

PONE-D-20-16666

Chest CT Findings Related to Mortality of Patients with COVID-19: A Retrospective Case-series Study

PLOS ONE

Dear Dr. Tao Ai,

Thank you for submitting your manuscript to PLOS ONE. After careful consideration, we feel that it has merit but does not fully meet PLOS ONE’s publication criteria as it currently stands. Therefore, we invite you to submit a revised version of the manuscript that addresses the points raised during the review process.

I have received the comments of the reviewers on your manuscript. The specific comments of the reviewers are included below. Please provide point by point response in your revised manuscript.

We look forward to receiving your revised manuscript.

Kind regards,

Muhammad Adrish

Academic Editor

PLOS ONE

Journal Requirements:

2. Please correct your reference to "p=0.000" to "p<0.001" or as similarly appropriate, as p values cannot equal zero.

4. Thank you for stating the following in the Financial Disclosure section:

'The author(s) received no specific funding for this work.'

We note that one or more of the authors are employed by a commercial company: Julei Technology Company

Reviewers' comments:

Reviewer's Responses to Questions

**Comments to the Author**

1. Is the manuscript technically sound, and do the data support the conclusions?

Reviewer #1: Yes

Reviewer #2: Partly

2. Has the statistical analysis been performed appropriately and rigorously? 

Reviewer #1: Yes

Reviewer #2: Yes

3. Have the authors made all data underlying the findings in their manuscript fully available?

Reviewer #1: Yes

Reviewer #2: Yes

4. Is the manuscript presented in an intelligible fashion and written in standard English?

Reviewer #1: Yes

Reviewer #2: No

5. Review Comments to the Author

Reviewer #1: I read with interest the mansuscript. I find it well wrote.

Only some suggestions:

1. Introduction: -include data on COVID 19 global burden at the time of the revision

2. Methods: clear

3. Results: I appreciate a lot the results, the table and the figure. Well done!

4. Discussion: discuss about the future perspective from your data and cite this article ( Coronavirus Diseases (COVID-19) Current Status and Future Perspectives: A Narrative Review. Int J Environ Res Public Health. 2020;17(8):2690. Published 2020 Apr 14.)

Reviewer #2: The manuscript entitled:"Chest CT Findings Related to Mortality of Patients with COVID-19: A Retrospective Case-series Study" is a retrospective study about the correllation between chest CT scan features, inflammatory serological markers and mortality rate in COVID-19 patients.

The questions are:

1) The Author reports an increase in neutrophyl/limphocites ratio near the death but does not specify the number of patients who had bacterial infection, can clarify please?

2) The Author reports the severity score of chest CT scan, can clarify the meaning of " severity score"?

3) What are the clinical signs of disease worsening as intended by the Author?

4) Why did the Author chose to measure the receptor of IL-2 as inflammatory marker?

6. PLOS authors have the option to publish the peer review history of their article (what does this mean?). If published, this will include your full peer review and any attached files.

Reviewer #1: **Yes: **Francesco Di Gennaro

Reviewer #2: No

---

## [Author Response · Author response to Decision Letter 0]

23 Jul 2020

Dear Dr. Muhammad Adrish (Academic Editor) 

We appreciate the timely and thoughtful comments from you and the reviewers on our manuscript (PONE-D-20-16666) entitled "Chest CT Findings Related to Mortality of Patients with COVID-19: A Retrospective Case-series Study”. We thoroughly reviewed all comments and have carefully revised our manuscript accordingly. 

Attached is our revised manuscript and detailed responses to the comments point by point. We hope that you will find our changes satisfactory. 

Again, thank you very much and we look forward to hearing from you soon!

Sincerely,

TAO AI

Journal Requirements:

Response: : Truly thank you for the comment. We have modified the article to meet PLOS ONE’s style requirements.

2.Please correct your reference to "p=0.000" to "p<0.001" or as similarly appropriate, as p values cannot equal zero.

Response: : Thank you for the comment. We have modified the P value in the article.

3.Thank you for stating the following in the Financial Disclosure section:

'The author(s) received no specific funding for this work.'

We note that one or more of the authors are employed by a commercial company: Julei Technology Company

Response: : Thank you for the comment. We decided to delete the author from our article, and thanks his contribution in Acknowledgement.

Comments to the Author:

Reviewer #1

Introduction: -include data on COVID 19 global burden at the time of the revision

Response: : Truly thank you for the comment. The related information has been updated.

Methods: clear

Response: Thanks for your comment. 

Results: I appreciate a lot the results, the table and the figure. Well done!

Response: Thanks for your comment.

Discussion: discuss about the future perspective from your data and cite this article ( Coronavirus Diseases (COVID-19) Current Status and Future Perspectives: A Narrative Review. Int J Environ Res Public Health. 2020;17(8):2690. Published 2020 Apr 14.)

Response: Thanks for your comment. We discussed about the future perspective from your data and added the literature according your suggestion.

Reviewer #2

The Author reports an increase in neutrophyl/lymphocites ratio near the death but does not specify the number of patients who had bacterial infection, can clarify please?

Response: Thanks for your concern. Yes，elevated Neutrophils, high sensitive CRP and decreased Lymphocytes were very common in patients with severe COVID-19, suggesting that second pulmonary bacterial infection plays an important role in the progression of the diseases (which may correlated with rapidly progressing lung consolidation/white lung). However, pathogen identification such as bacteria culture to confirm the infection of bacteria had been performed during the early time of the outbreak because of the long-time consumption of bacteria culture and potential risks when obtaining sputum specimens. Thus, in Chinese practice, early empirical treatment with antibiotics covering common pathogens was strongly suggested to be administered after analyzing clinical symptoms, and marked elevated levels of neutrophils and hs-CRP in serum, to prevent the occurrence of septic sepsis/multi-organ dysfunction . 

The Author reports the severity score of chest CT scan, can clarify the meaning of " severity score"?

Response: Thanks for your comment. Severity score of chest CT means the range of lesions involvement. The higher the score, the wider the range of lesion involvement. The relevant content was mentioned in article, which appears in Materials and methods section CT image analysis: “Each of the five lung lobes was visually scored for the degree of lung involvements using a 4-point- scale: 0, no involvement; 1, 1-25% involvement; 2, 26%-49% involvement; 3, 50%-75% involvement; 4, 76%-100% involvement. The total severity CT score (the extent of pulmonary disease) was the sum of the five individual lobar scores and defined as follows: 0, none; 1-5, minimal; 6-10, mild; 11-15, moderate; and 16-20, severe involvement of the lung (white lung).” The method of severity score evaluated chest CT was from Bernheim [1].

[1] Bernheim A, Mei X, Huang M, et al. Chest CT Findings in Coronavirus Disease-19 (COVID-19): Relationship to Duration of Infection. Radiology. 2020;295(3):200463.

What are the clinical signs of disease worsening as intended by the Author?

Response: Thanks for your comment. In clinical aspect, the signs of disease worsening refers dyspnea that cannot be improved, irreversibly decreased blood oxygen saturation, decreased lymphocytes and increased level of inflammatory factors. In radiology aspect, the clinical signs of disease worsening are increased proportions of consolidation and increased chest CT severity scores.

Why did the Author choose to measure the receptor of IL-2 as inflammatory marker?

Response: Thanks for your comment. IL-2R, similar to TNF-α, IL-1β, IL-6 and IL-8, is a kind of inflammatory cytokines, which is a necessary signal for immune response. Meanwhile, pervious studies have demonstrated that the concentration of IL-2R was significantly changed in COVID-19 patients [2,3]. Therefore, we chose IL-2R.

[2] Chen G, Wu D, Guo W, et al. Clinical and immunological features of severe and moderate coronavirus disease 2019. J Clin Invest. 2020;130(5):2620-2629. 

[3] Hou H, Zhang B, Huang H, et al. Using IL-2R/lymphocytes for predicting the clinical progression of patients with COVID-19. Clin Exp Immunol. 2020;201(1):76-84.

---

## [Decision Letter · Decision Letter 1]

27 Jul 2020

Chest CT Findings Related to Mortality of Patients with COVID-19 : A Retrospective Case-series Study

PONE-D-20-16666R1

Dear Dr. Tao Ai,

We’re pleased to inform you that your manuscript has been judged scientifically suitable for publication and will be formally accepted for publication once it meets all outstanding technical requirements.

Kind regards,

Muhammad Adrish

Academic Editor

PLOS ONE

Additional Editor Comments (optional):

Reviewers' comments:

Reviewer's Responses to Questions

**Comments to the Author**

1. If the authors have adequately addressed your comments raised in a previous round of review and you feel that this manuscript is now acceptable for publication, you may indicate that here to bypass the “Comments to the Author” section, enter your conflict of interest statement in the “Confidential to Editor” section, and submit your "Accept" recommendation.

Reviewer #1: All comments have been addressed

Reviewer #2: All comments have been addressed

2. Is the manuscript technically sound, and do the data support the conclusions?

Reviewer #1: Yes

Reviewer #2: Yes

3. Has the statistical analysis been performed appropriately and rigorously? 

Reviewer #1: Yes

Reviewer #2: Yes

4. Have the authors made all data underlying the findings in their manuscript fully available?

Reviewer #1: Yes

Reviewer #2: Yes

5. Is the manuscript presented in an intelligible fashion and written in standard English?

Reviewer #1: Yes

Reviewer #2: Yes

6. Review Comments to the Author

Reviewer #1: I appreciate a lot your paper

Authors improve their manuscript following reviewer suggestions

I think that is a good example of good interaction beetwen editor authors and reviewer

Congratulations

Reviewer #2: The Author replied satisfactorily to all questions and in my opinion the paper can be published in PlosOne journal

7. PLOS authors have the option to publish the peer review history of their article (what does this mean?). If published, this will include your full peer review and any attached files.

Reviewer #1: **Yes: **Francesco Di Gennaro

Reviewer #2: No

---

## [Editor Report · Acceptance letter]

17 Aug 2020

PONE-D-20-16666R1 

Chest CT Findings Related to Mortality of Patients with COVID-19: A Retrospective Case-series Study 

Dear Dr. Ai:

I'm pleased to inform you that your manuscript has been deemed suitable for publication in PLOS ONE. Congratulations! Your manuscript is now with our production department. 

Kind regards, 

on behalf of

Dr. Muhammad Adrish 

Academic Editor

PLOS ONE